# Comparative Analysis of the GATA Transcription Factors in Five Solanaceae Species and Their Responses to Salt Stress in Wolfberry (*Lycium barbarum* L.)

**DOI:** 10.3390/genes14101943

**Published:** 2023-10-15

**Authors:** Fengfeng Zhang, Yan Wu, Xin Shi, Xiaojing Wang, Yue Yin

**Affiliations:** 1Institute of Quality Standards and Testing Technology for Agricultural Products, Ningxia Academy of Agricultural and Forestry Sciences, Yinchuan 750002, China; zhfeng99998@163.com (F.Z.); wfsnow_1985@163.com (Y.W.); sxdewp@163.com (X.S.); 2National Wolfberry Engineering Research Center, Ningxia Academy of Agricultural and Forestry Sciences, Yinchuan 750002, China

**Keywords:** wolfberry, GATA gene family, Solanaceae, salt stress, gene expression

## Abstract

GATA proteins are a class of zinc-finger DNA-binding proteins that participate in diverse regulatory processes in plants, including the development processes and responses to environmental stresses. However, a comprehensive analysis of the GATA gene family has not been performed in a wolfberry (*Lycium barbarum* L.) or other Solanaceae species. There are 156 *GATA* genes identified in five Solanaceae species (*Lycium barbarum* L., *Solanum lycopersicum* L., *Capsicum annuum* L., *Solanum tuberosum* L., and *Solanum melongena* L.) in this study. Based on their phylogeny, they can be categorized into four subfamilies (I-IV). Noticeably, synteny analysis revealed that dispersed- and whole-genome duplication contributed to the expansion of the GATA gene family. Purifying selection was a major force driving the evolution of *GATA* genes. Moreover, the predicted *cis*-elements revealed the potential roles of wolfberry *GATA* genes in phytohormone, development, and stress responses. Furthermore, the RNA-seq analysis identified 31 *LbaGATA* genes with different transcript profiling under salt stress. Nine candidate genes were then selected for further verification using quantitative real-time PCR. The results revealed that four candidate *LbaGATA* genes (*LbaGATA8*, *LbaGATA19*, *LbaGATA20*, and *LbaGATA24*) are potentially involved in salt-stress responses. In conclusion, this study contributes significantly to our understanding of the evolution and function of *GATA* genes among the Solanaceae species, including wolfberry.

## 1. Introduction

There are several adaptive mechanisms that plants have developed to cope with these abiotic stresses, including drought, salinity, and extreme temperatures during their growth and development [1]. To adapt to these extreme environmental stresses, complex and efficient regulatory networks, including changes in gene expression and transcription factors (TFs), are critical regulators of the plant’s adaptation to abiotic stresses [2]. Several families of TFs have been identified and reported in response to abiotic stresses, such as GATA-binding factor (GATA), V-myb avian myeloblastosis viral oncogene homolog (MYB), WRKY, Basic region leucine zipper (bZIP), GRAS (GAI, RGA, and SCR), NAC (NAM, ATAF1/ATAF2, and CUC2), Dehydration-responsive element-binding protein (DREB), and Basic helix–loop–helix (bHLH) [3,4,5,6,7,8,9,10]. However, only a few TFs have been identified and analyzed in *Lycium barbarum* L. and *Lycium ruthenicum* M., including R2R3-MYB, BBX, and WRKY [11,12,13]. Thus, the role of other crucial TFs, such as GATA, and the function of these proteins in *L. barbarum* are unclear.

GATA TFs are DNA-binding proteins widely distributed in eukaryotes, including fungi, metazoans, and plants [14]. They are a type IV zinc finger TF family member, with the conserved motif sequence C-X2C-X17–20-C-X2−C followed by a basic region. The GATA TFs recognize and bind the WGATAR (W = T or A; R = G or A) sequences regulating the transcription levels of their downstream genes [15]. In plants, most GATA proteins contain a single zinc-finger domain, C-X2-C-X18-C-X2-C, with a few proteins containing two zinc-finger domains, C-X2-C-X20-C-X2-C [16]. Based on the phylogenetic analysis, DNA-binding domains, and gene structures of *Arabidopsis thaliana*, the *GATA* family was divided into subfamilies I, II, III, and IV.

The GATA TFs have been found to play a critical role in regulating plant growth, development, and response to abiotic stresses in previous studies. For instance, in *Arabidopsis thaliana*, ectopic overexpression of either *AtGNC* or *AtCGA1* promotes chloroplast biogenesis in the hypocotyl cortex and root pericycle cells [17]. In soybean, the overexpression of the *GmGATA58* increases the chlorophyll content in the leaves but suppresses plant growth and yield in transgenic *Arabidopsis* [17]. In *Brachypodium distachyon*, the overexpression of the *BdGATA13* in *Arabidopsis* results in darker green leaves, late flowering, and increased drought tolerance compared to the wild type [18]. In tomato, the overexpression of the *SlGATA17* improves drought resistance by regulating the activity of the phenylpropanoid biosynthesis pathway [19]. Additionally, under salt stress, rice Os*GATA8*-overexpressing lines have higher light efficiency and biomass than wild-type and mutant plants [20]. More importantly, *OsGATA16* is responsive to cold stress [21]. In sweet potato, the overexpression of the *IbGATA24*, a member of the subfamily III, significantly increases resistance to drought and salt stresses by interacting with the COP9-5a protein [22]. Additionally, the overexpression of *TaGATA1* significantly enhances wheat resistance to *Rhizoctonia cerealis*, whereas silencing *TaGATA1* suppresses the resistance [23]. The GATA transcription factors play an important role in nitrogen metabolism as well. For example, the overexpression of *PdGNC*, an ortholog of *AtGNC* in *Arabidopsis*, increases starch accumulation and promotes plant growth under nitrogen deficiency [24]. In general, *GATA* genes play an important role in many biological processes.

The *GATA* gene has been identified in several plant species, including 29 in *Arabidopsis* [3], 79 in wheat [25], 49 in potato [26], 28 in foxtail millet [27], 28 in buckwheat [28], 26 in cucumber [29], and 32 in Chinese pear [30]. To date, five species from the Solanaceae family have their whole genome sequenced and deposited in the Sol Genomics Network (SGN) database [31] and Spud DB [32]. Recently, the genome sequencing of wolfberry (*Lycium barbarum* cv. ‘Ningqi 1′) has been completed [33]. These genomic resources are informative for comparative analyses of the GATA gene family among the Solanaceae family. 

Therefore, this study aimed to explore the potential regulatory roles of *LbaGATA* genes in response to salt stress. A genome-wide identification and comparative analysis of the *GATA* gene family was performed in five species, including wolfberry (*Lycium barbarum* L.), tomato (*Solanum lycopersicum* L.), pepper (*Capsicum annuum* L.), potato (*Solanum tuberosum* L.), and eggplant (*Solanum melongena* L.) from the Solanaceae family. In addition, a comprehensive phylogenetic analysis, exon–intron structure, motif composition, gene duplication, and *cis*-acting element analysis were performed. Moreover, the responses of *GATA* family members to salt stress in wolfberry were determined by combining transcriptome and qRT-PCR analyses. Based on these analyses, 156 *GATA* genes were identified. As a result of this study, we will gain a better understanding of the evolutionary relationship among the *GATA* genes in the family Solanaceae. Moreover, they will be useful to further characterize the functions of *GATA* genes in wolfberry and other Solanaceae plants.

## 2. Materials and Methods

### 2.1. Plant Materials and Salt Stress

Wolfberry (*L. barbarum* cv Ningqi 1) seeds were sown on moist filter papers on Petri plates for germination. Next, the seedlings were transplanted into a plastic box containing 1/2 Hoagland solution and maintained for four weeks. Subsequently, the seedlings were subjected to salt treatments. Briefly, the seedlings were transferred into a hydroponic culture consisting of a Hoagland solution under a 300 mM NaCl treatment and incubated at 22/18 °C (day/night), 65% relative humidity, and a 16 h photoperiod. Subsequently, the leaves’ tissues were collected at 0 h (samples without treatment, CK), 1 h, 3 h, 6 h, 12 h, and 24 h after salt application, grounded, frozen in liquid nitrogen, and stored at −80 °C awaiting RNA-seq and qRT-PCR analyses. There were at least six seedlings and three biological replicates in each treatment.

### 2.2. Identification and Characterization of the GATA Genes in Five Solanaceae Family Members

Annotations and genome sequences of *Arabidopsis* and five members of the Solanaceae family, including *L. barbarum*, *S. lycopersicum*, *C. annuum*, *S. tuberosum*, and *S. melongena*, were obtained from online databases. Precisely, the wolfberry genome data (accession number PRJNA640228) were downloaded from the NCBI [33]. The genome sequences of the other four species were downloaded from the SGN (https://solgenomics.net/, accessed on 10 October 2022) and Spud DB (http://spuddb.uga.edu/index.shtml, accessed on 10 October 2022). The *A. thaliana* genome data were retrieved from the Arabidopsis Information Resource (https://www.arabidopsis.org/, accessed on 10 October 2022).

The *GATA* genes in the five Solanaceae family members were identified using two methods as previously described [12]. First, the 29 *Arabidopsis* AtGATA protein sequences were used as queries for the genome data of the five species in the Solanaceae family using the BLASTP Program (E-value < 10^−5^). Second, the Pfam database (http://pfam.xfam.org/, accessed on 10 October 2022) was used to retrieve the genome data of the five species using the Hidden Markov Model (HMM) of the GATA zinc finger (PF00032). Subsequently, the potential *GATA* genes in the five species genomes were further screened for the GATA domain against the Pfam (http://pfam.xfam.org/, accessed on 10 October 2022), SMART (http://smart.embl.de/, accessed on 10 October 2022), and batch Conserved Domain Search (https://www.ncbi.nlm.nih.gov/Structure/cdd/wrpsb.cgi, accessed on 10 October 2022) databases [34,35,36]. In addition, the ExPASy server (https://web.expasy.org/protparam/, accessed on 10 October 2022) predicted the physical and chemical parameters of the GATA proteins in the five species [37], including molecular weights (MW), isoelectric point (pI), and grand average of hydropathicity (GRAVY). Finally, GATA protein subcellular locations were predicted using WOLF-PSORT (https://wolfpsort.hgc.jp/, accessed on 12 October 2022) [38].

### 2.3. Phylogenetic Analysis of GATA Proteins in Five Species of Solanaceae Family and Arabidopsis

Multiple sequence alignment of the full-length GATA proteins was performed using Muscle v5 software with default parameters [39]. Next, the Maximum Likelihood (ML) tree was constructed using the IQ-TREE v2 software with the JTT + G substitution model [40]. The bootstrap test was carried out with 1000 iterations, using pairwise deletion of gaps. Finally, we visualized the phylogenetic tree using iTOL v5 (https://itol.embl.de/, accessed on 12 November 2022) [41].

### 2.4. Gene Features and Conserved Motifs of the GATA Gene Family

The *GATA* exons and introns were identified using the five Solanaceae genome annotation files. First, the Gene Structure Display Server (http://gsds.gao-lab.org/index.php, on 18 November 2022) [42] was used to visualize the exon–intron. Second, the full-length sequence of GATA proteins was submitted to the MEME suite (https://meme-suite.org/meme/, on 18 November 2022), and the conserved motifs were analyzed with a maximum of 20 different motifs and an optimal width of 6–50 bp [43].

### 2.5. Analysis of GATA Chromosomal Location, Gene Duplication Events, and Gene Synteny

The information on the chromosomal locations of the *GATA* genes in *L. barbarum*, *S. lycopersicum*, *S. melongena*, *S. tuberosum,* and *C. annuum* was obtained from their genome annotation files. Next, their chromosomal locations were mapped using the TBtools v1.12 [44]. Subsequently, the gene duplication events in genes derived from dispersed duplicates (DSD), proximal duplicates (PD), whole-genome duplicates (WGD), transposed duplicates (TRD), and tandem duplicates (TD) were investigated using the DupGen_finder pipeline, with *Arabidopsis* as the outgroup. [45]. Finally, the TBtools v1.12 [44] performed syntenic analysis to reveal the syntenic relationships between LbaGATA and GATA proteins from *S. lycopersicum*, *C. annuum*, *S. tuberosum*, *S. melongena*, and *A. thaliana*.

### 2.6. Nonsynonymous (Ka) and Synonymous (Ks) Calculation

Ka, Ks, and Ka/Ks substitution rates were calculated using the Nei–Gojobori (NG) method in TBtools v1.12, using the coding sequences, protein sequences, and syntenic gene pairs [44,46].

### 2.7. Promoter Sequence Analysis of LbaGATA Genes

Based on the wolfberry genome database, the upstream sequences (2.0 kb) from the initiation codon (ATG) of *LbaGATAs* were extracted using TBtools v1.12 [44] to examine their *cis*-regulatory elements. PlantCARE was used to predict the *cis*-acting elements in the promoter sequences [47].

### 2.8. RNA-seq Analysis

The ground-above tissues at 0, 1, 3, 6, 12, and 24 after salt stress treatments were collected and applied to RNA-seq analysis. Total RNA was extracted from treatment samples using the Trizol method as described previously [12], and mRNA libraries were sequenced on the Illumina HiSeq4000 platform. The raw data were filtered to obtain clean reads. Subsequently, the cleaned reads were mapped to the wolfberry reference genome using HISAT2 v2.05 [48]. Next, the differentially expressed genes (DEGs) between the mapped sequences and the reference genome were identified using the DESeq2 v1.18 [49] with |log_2_(fold change)| ≥ 1 and false discovery rate < 0.05 parameters. The Kyoto Encyclopedia of Genes and Genomes (KEGG) pathways enriched with the DEGs were detected with KOBAS v3.0 [50]. There was a significant enrichment of KEGG pathways with a *q*_value of 0.05. Subsequently, RNA-seq data were used to extract the transcript profiling data of *LbaGATA* genes, and the heatmap was generated with the corresponding fragments per kilobase of transcript per million mapped reads (FPKM) using TBtools v1.12 [44]. 

### 2.9. Quantitative Real-Time PCR Analysis

The qRT-PCR was used to analyze the expression patterns of the *LbaGATA* genes. Total RNA was extracted from leaves (the sample same as transcriptome analysis) using Trizol reagent (TIANGEN, Beijing, China) following the manufacturer’s instructions. The first-strand cDNA was synthesized using EasyScript One-Step gDNA Removal and cDNA Synthesis SuperMix (TransGen, Beijing, China) according to the manufacturer’s instructions. Next, qRT-PCR was performed on the Bio-Rad CFX96 Touch^TM^ Real-Time PCR detection system (Bio-Rad, Foster City, California, USA). qRT-PCR was performed in a total volume of 15 μL, including 2 μL of cDNA template, 0.6 μL of 10 μM primer mixture, 7.5 μL Mix (TransGen, Beijing, China), and 4.9 μL RNase-free water. The amplification conditions were as follows: 95 °C for 3 min, 40 cycles of 95 °C for 10 s, and 58 °C for 30 s. *Actin* was used as the reference gene [51]. As shown in Appendix A, Primer Premier 5.0 software was used to design gene-specific primers for each *LbaGATA* gene. Finally, the relative gene expression was calculated using the 2^−∆∆Ct^ method [52]. An independent sample *t*-test was used to analyze the statistical significance of the data.

## 3. Results

### 3.1. Identification and Characterization of the GATA Genes in the Five Solanaceae Species

A total of 159 candidate *GATA* genes were retrieved from the genomes of the five species. Among them, three sequences lacking the GATA-type zinc-finger motifs were removed. Finally, a total of 156 *GATA* genes were found in wolfberry (31), tomato (32), pepper (28), potato (36), and eggplant (29) (Table 1). The length of the encoded GATA proteins was 103–955 aa with an average of 309 amino acids across the five species (Appendix A and Figure 1). The GATA protein molecular weight and pI were between 11.434 and 102.734 kDa with an average of 34.232 kDa and between 4.41 and 10.13 with an average of 7.62, respectively (Appendix A). All proteins had negative GRAVY values, implying that they were hydrophilic proteins, only differing in their degree of hydrophilicity. Most *GATA* genes were found in the nucleus, with a few in the cytoplasm and chloroplast, suggesting that GATAs are nuclear-localized transcription factors.

### 3.2. Phylogenetic Analysis of GATA Proteins in the Five Species from the Solanaceae Family and Arabidopsis

All the 186 GATA proteins, including 31 in wolfberry, 32 in tomato, 28 in pepper, 36 in potato, 29 in eggplant, and 30 in *Arabidopsis,* were grouped into four subfamilies (Figure 1). Subfamily I was the largest, comprising 101 GATA proteins, followed by 55, 23, and 7 GATA proteins in subfamily II, III, and IV, respectively. The phylogenetic tree had an obvious topology, with proteins in each subfamily clustered in a single branch. The findings demonstrate evolutionarily conserved *GATA* genes and stronger homology between them when evaluated synchronously.

### 3.3. Gene Features and Conserved Motifs of the GATA Gene Family

The intron–exon structure and conserved motifs of GATA proteins in five Solanaceae species were analyzed. The Maximum Likelihood phylogenetic tree was constructed using the full-length GATA proteins (Figure 2a and Appendix A). *LbaGATA* genes contained 2–19 exons, indicating a wide range of variation (Figure 2b). Among the 31 *LbaGATA* genes in wolfberry, 14 had two exons. Furthermore, the structure of *LbaGATA* genes was relatively conserved with the same subfamily but varied across different subfamilies. The number of exons was higher for genes belonging to subfamily III and IV and lower for genes in subfamily I and II, except for *LbaGATA28* (19).

The conserved LbaGATA protein motifs highlighting their evolutionary relationship and classification are presented in Figure 2c. Among these LbaGATA proteins, there were 20 motifs with 11–50 amino acids in each motif (Appendix A). Additionally, nearly all the LbaGATA proteins in the same subfamily had relatively conserved motif composition and arrangement. However, there was a substantial degree of variation in the different subfamilies.

It was found that all LbaGATA proteins contained conserved motifs 1 and 3, except LbaGATA2, which lacked the conserved Motif 3. Precisely, subfamily I contained Motifs 2, 4, 8, and 9, while Motifs 10, 12, and 19 were found only in subfamily II. However, the LbaGATA protein motif composition and arrangement in subfamily III were highly conserved, with only LbaGATA16 lacking Motifs 11 and 3 and LbaGATA18 lacking motif 11. Additionally, the LbaGATA proteins in subfamily IV generally contained more unique Motifs, including Motifs 5, 11, 15, 16, and 17. LbaGATA8 lacked Motif 8 (special to subfamily I and IV) compared to LbaGATA13 and LbaGATA23, which have identical Motif arrangements.

### 3.4. Gene Duplication Events, Chromosomal Location, and Synteny Analysis of GATA Genes

Further comprehensive analysis of gene duplication events in five Solanaceae species, was conducted, including various types such as whole-genome duplication (WGD), proximal duplication (PD), tandem duplication (TD), transposed duplication (TRD), and dispersed duplication (DSD). The software DupGen_finder v1.10 [45] was used to identify duplicate *GATA* family gene pairs within five genomes of Solanaceae. The WGD, TD, TRD, and DSD duplication events were present in all the five species from the Solanaceae family, while PD duplication event was traced only in tomato, pepper, and potato. There were 249 duplicate genes in the five species, with DSDs having the largest number of gene pairs (132), followed by WGDs with 73, and TRDs with 28. In contrast, only 24 TD and 3 PD pairs were identified (Figure 3). These results imply that the expansion of the *GATA* gene family was mainly associated with DSD and WGD events. The WGD pairs in wolfberry (17), tomato (18), potato (16), and eggplant (14) were greater than those in pepper (7), suggesting that WGDs play important roles in the *GATA* family expansion in wolfberry, tomato, potato, and eggplant (Appendix A).

Additionally, most *GATA* genes were unevenly distributed at both ends of the chromosomes (Figure 4). For wolfberries, 31 *GATA* genes were located on 11 of the 12 chromosomes. There were seven genes distributed on Chr6, which contained most genes compared to other chromosomes, while Chr7 and Chr10 contained only one *GATA* gene (Figure 4a). In tomato, there were two clusters of homologous *GATA* genes on Chr1 and Chr5 (Figure 4b). In pepper, there were three clusters of homologous *GATA* genes on Chr2, Chr5, and Chr8, with three to four genes per cluster (Figure 4c). In potato, there were two clusters of *GATA* genes on Chr1 and Chr5, each with three to twelve genes (Figure 4d). In eggplant, two *GATA* gene clusters were identified on Chr1 and Chr5, with three to five genes per cluster (Figure 4e).

Further intra-genomic collinearity revealed 17 gene pairs in wolfberry, 18 in tomato, 6 in pepper, 18 in potato, and 14 in eggplant (Figure 5 and Appendix A). As a result, 270 orthologous gene pairs were identified among the five species based on collinearity analyses of the *GATA* genes (Figure 5 and Appendix A), including 80, 26, 60, 87, and 17 collinear gene pairs between wolfberry and tomato, pepper, potato, eggplant, and *Arabidopsis*, respectively. These results imply that wolfberry and the other four species in the Solanaceae species have a good collinearity relationship, which suggests a possible evolutionary mechanism between them.

### 3.5. GATA Genes Evolved under Strong Purifying Selection

In Figure 6 and Appendix A, paralogous *GATA* gene pairs are shown as non-synonymous (Ka), synonymous (Ks), and Ka/Ks. The Ka/Ks ratios of duplicated gene pairs in wolfberry, tomato, pepper, and eggplant were <1, implying that *GATA* genes evolved under strong purifying selection. However, in potatoes, one DSD gene pairs *Soltu.DM.01G043530.1* and *Soltu.DM.01G047500.2* (Ka/Ks~1.06) had higher Ka/Ks ratios, implying that this family has a complicated evolutionary history. In wolfberry, the calculated mean Ka/Ks values for WGD, DSD, TD, and TRD gene pairs were 0.22, 0.23, 0.25, and 0.14, respectively (Figure 6 and Appendix A). In comparison with the other duplicated gene pairs, the DSD gene pairs had a higher Ka/Ks ratio, indicating a higher rate of evolution (Figure 6a). 

### 3.6. Analysis of LbaGATA Promoter Element Cis-Acting Elements

The *LbaGATA* gene expression regulation mechanism in the 2000 bp upstream sequences in the promoter regions of the 31 *LbaGATA* genes is shown in Figure 7a. In total, 14 *cis*-regulatory elements were identified and grouped into three basic physiological subgroups, including abiotic/biotic stress responsiveness, phytohormones responsiveness, and plant growth and development (Figure 7b). For plant growth and development, the 31 GATA box promoter sequences contained MYB, which was involved in various developmental processes/stresses. In addition, most of the *LbaGATA* family contained ARE *cis*-acting elements for anaerobic induction. The subgroup included TGACG-motif, CGTCA-motif, and ABRE, which regulate MeJA response and drought inducibility in phytohormones responsiveness. The members belonged to Subfamily I with high levels of various *cis*-acting elements, especially *LbaGATA13*, with most phytohormones related to elements like ABRE and plant growth related to elements like G-box (light responsiveness). However, only *LbaGATA19* had AuxRR-core (auxin responsiveness). Meanwhile, *LbaGATA4*, *LbaGATA18,* and *LbaGATA28* had the GARE motif related to gibberellin responsiveness.

### 3.7. Role of LbaGATA Family in Response to Salt Stress

To conduct transcript profiling under salt stress, we harvested leaves’ tissues from wolfberry plants after salt treatment for 0 h (control, CK), 1 h, 3 h, 6 h, 12 h, and 72 h. These tissues were subsequently used for RNA-seq analysis. Transcriptome characteristics of biological replicates from the same treatment (Appendix A) were highly correlated (r^2^ > 0.8), suggesting the reproducibility of transcriptome data. Therefore, these transcriptome data could be used for further analysis. A total of 1437, 3258, 4582, 4518, and 2903 genes were transcriptionally up-regulated or down-regulated at 1, 3, 6, 12, and 24 h after salt treatment, respectively (Figure 8a and Figure 8b). K-means clustering analysis revealed that all the genes were divided into 12 different clusters, which indicated that there were a lot of gene expression differences in different samples (Appendix A). KEGG analysis revealed that the altered genes, 1 h after salt treatment, were enriched in a plant–pathogen interaction, MAPK signaling pathway–plant, and plant hormone signal transduction (Figure 8c). After 3, 6, and 12 h of salt treatment, the enriched genes were involved in the MAPK signaling pathway, ribosomes, photosynthesis, antenna proteins, porphyrin metabolism, carbon fixation in photosynthetic organisms, and circadian rhythm (Figure 8d–f). After 24 h of salt treatment, the altered genes were mainly enriched in ribosomes (Figure 8g).

In order to conduct transcript profiling under salt stress, we extracted the transcriptomic data for the 31 *LbaGATA* genes. Consequently, 31 *LbaGATA* genes showed different transcript profiles under salt stress (Figure 9). These genes were divided into three main groups based on their expression profiles. Group I contained 12 genes (*LbaGATA4/8/10/11/12/14//15/17/19/20/21/22/24*) (Figure 9 and Appendix A), of which 11 were increased with an increase in stress duration, especially *LbaGATA8*, *LbaGATA12*, *LbaGATA17*, *LbaGATA19*, and *LbaGATA24*, which increased by 2.34, 3.68, 2.07, 2.64, and 3.1-fold, respectively. In group II, the *LbaGATAs* were transcriptionally decreased at the early stress stages and increased after, such as *LbaGATA2/6/22/30*. In group III, after salt treatment, there was no significant change in gene expression (Figure 9 and Appendix A). 

### 3.8. qRT-PCR-Based Analyses of Gene Expression

In order to verify the results of RNA-seq, nine *LbaGATA* genes were selected at random, and their expression levels were determined using qRT-PCR to analyze their expression levels after salt treatment for 0 (CK), 1, 3, 6, 12, and 24 h. All nine *LbaGATA* genes were up-regulated under salt stress (Figure 10). The expression levels of *LbaGATA8*, *LbaGATA17*, *LbaGATA19*, *LbaGATA20,* and *LbaGATA24* were induced with salt stress at one or more time point/s, which was consistent with the RNA-seq analysis. *LbaGATA8* had the highest expression at 6 h, while *LbaGATA17*, *LbaGATA19*, *LbaGATA20*, and *LbaGATA24* had the highest expression pattern at 12 h compared to the control (0 h). The results suggest that these *LbaGATAs* may play an important role in the salt stress response of wolfberries.

## 4. Discussion

GATA TFs are a member of the zinc-finger proteins, widely identified and functionally characterized in a wide range of plant species [53]. Studies have shown that *GATA* genes are involved in regulating growth, chlorophyll synthesis, and environmental response. [17,19,54]. However, there is little information about this gene family in wolfberry, and there has not been a systematic study of *GATA* genes in Solanaceae.

*GATA* genes exist broadly in plants, with 5335 *GATA* genes from 165 plant species deposited in the PlantTFDB [55], including 40 *GATA* genes in kiwifruit, 54 in maize, 92 in soybean, and 6 in sugarcane. However, the sizes of *GATA* families vary across the different plant species. In this study, the composition of the GATA gene families was also diverse among the five Solanaceae species. Considering the numbers of *GATA* genes in wolfberry (31), tomato (32), pepper (28), potato (36), and eggplant (29). Generally, the number of *GATA* genes in potato exceeded the other four Solanaceae species. Additionally, the genome size across the five Solanaceae species varied from 0.785 (tomato) to 3.30 Gb (pepper). This implies that the number of *GATA* genes in a plant is independent of the genome size. A recent study revealed that wolfberry and other hitherto sequenced Solanaceous plants experienced a whole-genome triplication event [33]. However, *GATA* genes were identified in four Solanaceae species, except wolfberry. Hence, WGD may not be the major factor driving the expansion of the GATA family in wolfberry and the four other Solanaceae species. Additionally, the GATA genes from all five Solanaceae species were classified into four clades according to their phylogenetic tree, consistent with those of other plant species [56].

Gene duplication contributes to the increase in new functional genetic materials and the development of new species [57]. As a common phenomenon, there are multiple pathways for gene duplication events, and the accumulation of beneficial mutations is preserved through selective evolution [58,59]. Generally, gene duplications are categorized as WGD, PD, TD, DSD, and TRD, which are the primary driving forces behind gene family evolution [45]. Gene duplication patterns contribute differently to gene family expansion. For example, large gene families are commonly affected by DSD and WGD, such as pyrabactin resistance 1-like (*PYR/PYL*) [60] and *BAHD* acyltransferase [61]. Similarly, the duplication of the *GATA* gene has been widely reported [30]. In this study, there were five duplication events identified in five Solanaceae species, including a large number of DSD (50.98%), WGD (33.33%), TRD (11.76%), and TD (3.92%) events in wolfberry, while PD events were not detected. Additionally, the DSD events accounted for 54, 52.78, 53.96, and 53.06% of all the duplication events in tomato, pepper, potato, and eggplant, respectively. Therefore, DSD and WGD are identified as the primary models of *GATA* gene family expansion in the five Solanaceae species, similar to the *GATA* family in *Arabidopsis* [45]. However, the occurrence time of WGD and DSD events in homologous *GATA* pairs across the five Solanaceae species differed, implying that the evolutionary histories of *GATAs* are specific. Further, the Ka/Ks ratios of paralogous GATA gene pairs in wolfberry are generally less than 1, which indicates that they have undergone purifying selection. One pair of *GATA* paralogous genes in potato had undergone positive selection, suggesting that novel functions may have evolved.

Salt stress affects plant growth, resulting in a significant reduction in crop productivity [62]. Increasing evidence has revealed that *GATA* genes are involved in plant response to salt stress [22]. For example, the analysis of the *cis*-elements in the promoter regions of *CaGATA* genes revealed their potential regulation during salinity, heat, cold, and drought stresses in pepper [63]. Herein, several stress-related *cis*-elements were found in wolfberry, including those for ARE (*cis*-acting regulatory element essential for anaerobic induction), LTR (low-temperature responsiveness), MBS (MYB binding Site), and AuXRR-core (auxin responsiveness), suggesting that *GATA* genes potentially regulate drought and salt stress responses in wolfberry. In addition, the RNA-seq data revealed that 12 *LbaGATA* genes were transcriptionally altered with more than two-fold changes via salt stress, consistent with previous studies that identified four *GATA* genes in *Fagopyrum tataricum* and three *GATA* genes in *Brassica napus* [28,64]. Another four *GATA* genes, *SiGATA16*, *SiGATA18*, *SiGATA22*, and *SiGATA25,* were induced by NaCl in *Setaria italica* [27]. The altered expression patterns of *LbaGATA* genes suggest they have functional roles in response to salt stress. Further verification of whether *LbaGATAs* are involved in response to salt stress revealed that *LbaGATA14*, *LbaGATA15*, and *LbaGATA24* were highly expressed at 6, 3, and 12 h, respectively. In contrast, the expression levels of *LbaGATA8*, *LbaGATA17*, *LbaGATA19*, *LbaGATA20*, and *LbaGATA25* were always up-regulated under salt stress. Therefore, it is possible that *LbaGATA* genes contribute to salt stress resistance in wolfberries.

## 5. Conclusions

In this study, genome-wide identification and bioinformatic analysis of *GATA* genes were conducted in wolfberry (*L. barbarum*) and four other Solanaceae species (*S. lycopersicum*, *C. annuum*, *S. tuberosum*, and *S. melongena*). In total, 156 members of the *GATA* gene family were identified in the five Solanaceae genomes, of which 31 were found in wolfberry. There are four subfamilies in the *GATA* family (groups I-IV) based on phylogenetic analyses, conserved domain analyses, and gene structure analyses. DSD and WGD largely contribute to expanding the *GATA* gene family in the Solanaceae species. In addition, the purifying selection is the main evolutionary force imposed on *GATA* genes, except for a few gene pairs with Ka/Ks values greater than one. Overall, *LbaGATA8*, *LbaGATA17*, *LbaGATA19*, *LbaGATA20,* and *LbaGATA24* are involved in salt stress responses. The findings in this study will provide a foundation to understand better the molecular evolution of *GATA* genes in Solanaceae species and their molecular functions in wolfberry and other Solanaceae species.

## Figures and Tables

**Figure 1 genes-14-01943-f001:**
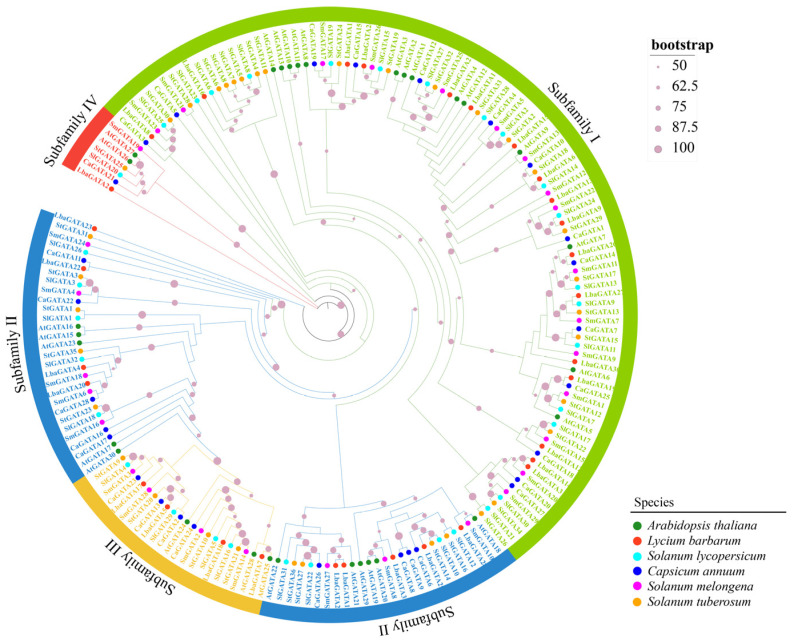
A phylogenetic analysis of GATA proteins in five Solanaceae species and Arabidopsis. The phylogenetic tree was constructed based on protein matrix using IQ-TREE v2 and grouped into four subfamilies (I–IV) with different colors. The bootstrap values are shown at each node. Green triangles, red rectangles, blue dots, gray triangles, yellow dots, and purple rectangles indicated the GATA proteins of Arabidopsis (AtGATA), wolfberry (LbaGATA), tomato (SlGATA), pepper (CaGATA), potato (StGATA), and eggplant (SmGATA), respectively.

**Figure 2 genes-14-01943-f002:**
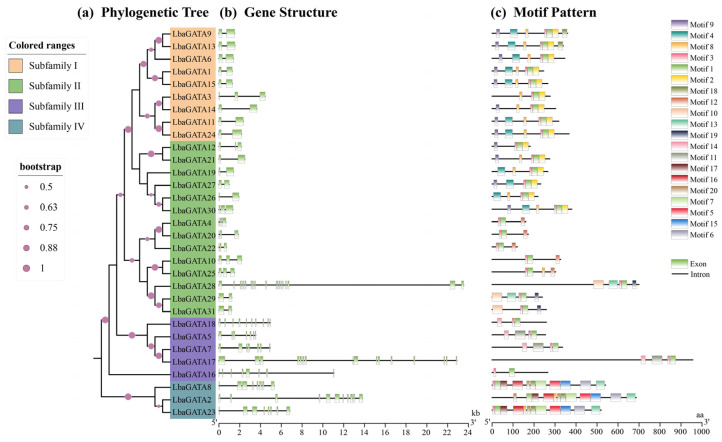
Phylogenetic relationship, conserved protein motifs, and gene structure in *LbaGATA* genes. (**a**) The Maximum Likelihood (ML) phylogeny includes 31 GATA proteins from wolfberry, grouped into 4 subfamilies, sequentially designated as Subfamily I to Subfamily IV. (**b**) Exon/intron structures of *LbaGATA* genes. Green boxes indicate exons, and introns are represented with black lines. A scale at the bottom can indicate the length of exons. (**c**) The motif composition of LbaGATA proteins. Twenty conserved motifs were performed using MEME, displayed in different colored boxes. The sequence information is shown in Appendix A. Protein length can be estimated with the scale at the bottom.

**Figure 3 genes-14-01943-f003:**
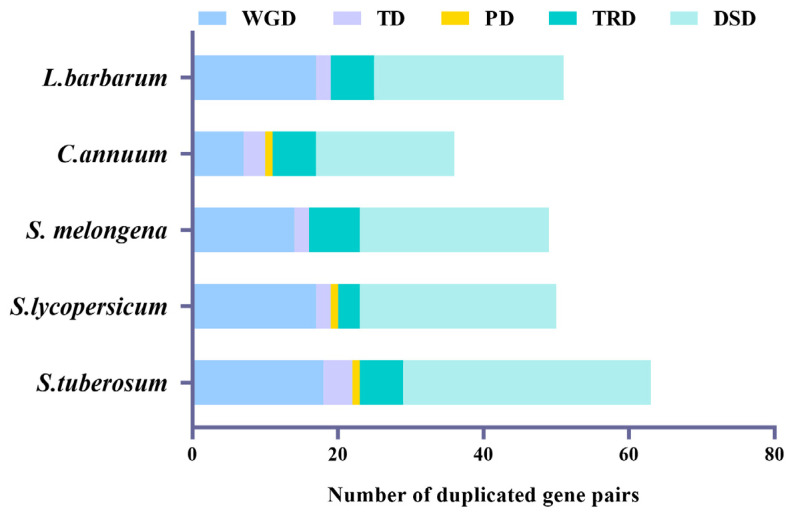
Gene duplications (TD, tandem duplicates; DSD, dispersed duplicates; WGD, whole-genome duplicates; PD, proximal duplicates; and TRD, transposed duplicates) of the GATA gene family in the five Solanaceae species. Different colored bars indicate different gene duplications.

**Figure 4 genes-14-01943-f004:**
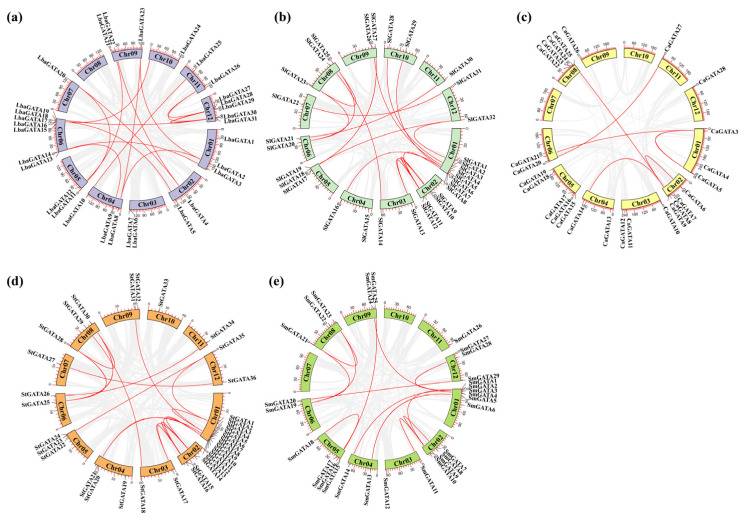
Syntenic relationship and chromosomal localization of the *GATA* gene family. (**a**) Wolfberry; (**b**) tomato; (**c**) pepper; (**d**) potato; and (**e**) eggplant. The different chromosomes were mapped with the *GATA* genes in the five Solanaceae species in different colors. Red line indicates the presence of duplicate gene pairs.

**Figure 5 genes-14-01943-f005:**
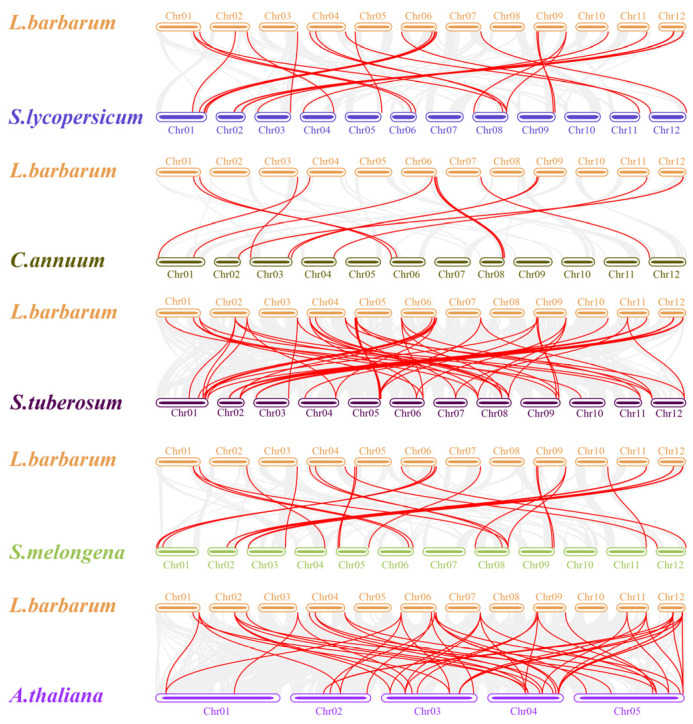
Synteny analysis of *GATA* genes in *Lycium barbarum* and five other plant species (*Solanum lycopersicum*, *Capsicum annuum*, *Solanum tuberosum*, *Solanum melongena*, and *Arabidopsis thaliana*). The gray lines represent collinear blocks in *L. barbarum* and the other plant genomes, while red lines highlight syntenic *GATA* gene pairs in *L. barbarum*.

**Figure 6 genes-14-01943-f006:**
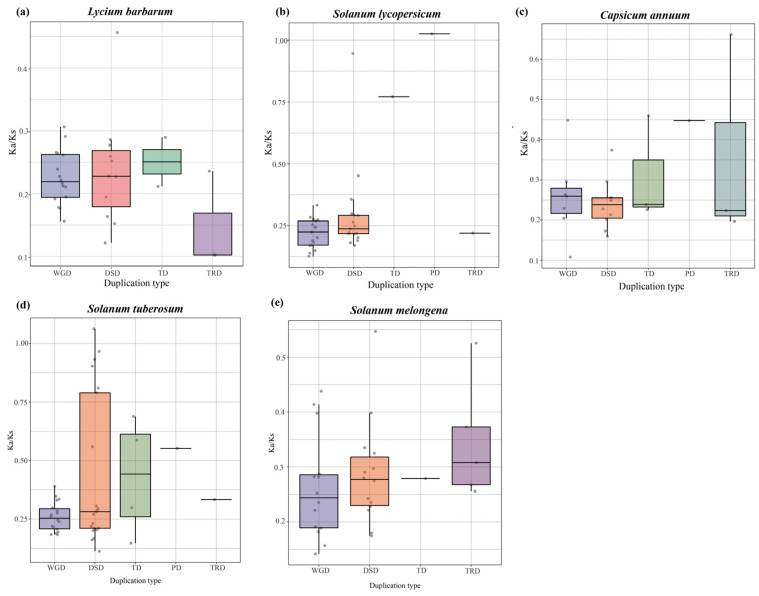
Analysis of the Ka/Ks values of *GATA* genes in five species of Solanaceae. Comparison of Ka/Ks values for different models of gene duplications. The x-axis represents five different duplication types. The y-axis indicates the Ka/Ks ratio. WGD, whole-genome duplicates; DSD, dispersed duplicates; TD, tandem duplicates; PD, proximal duplicates; and TRD, transposed duplicates. (**a**): wolfberry; (**b**): tomato; (**c**): pepper; (**d**): potato; (**e**): eggplant.

**Figure 7 genes-14-01943-f007:**
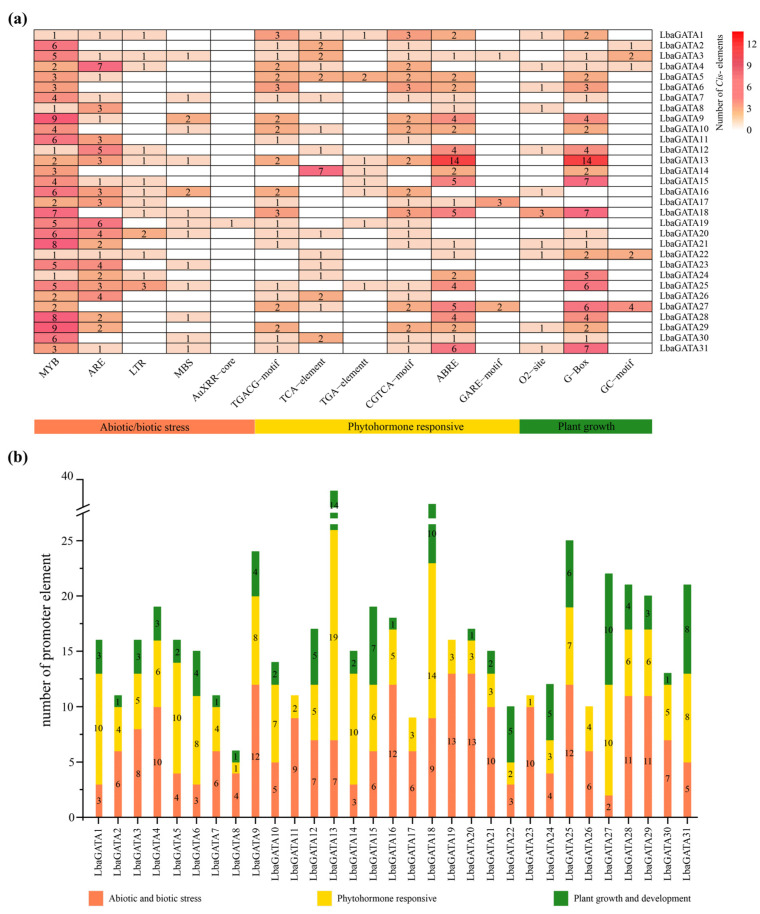
*Cis*-acting elements analysis in the promoter of *LbaGATA* Genes. (**a**) Special coding colors indicate *cis*-acting elements on the upstream regions of LbaGATA genes with functional similarity. (**b**) Abiotic/biotic stress, phytohormone responses, and plant growth development for *cis*-elements are represented using different colors.

**Figure 8 genes-14-01943-f008:**
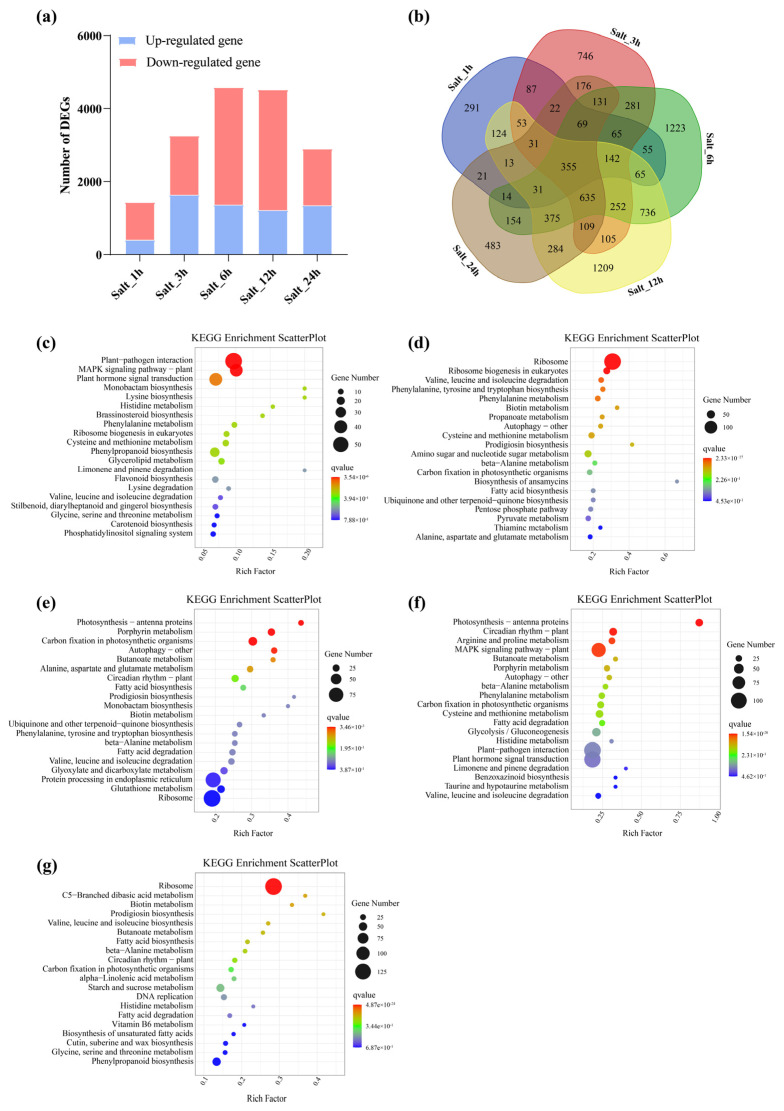
Transcriptomic analysis of wolfberry seedlings under salt stress. (**a**) Numbers and Venn diagram (**b**) of differentially expressed genes under salt treatment for 0 h (CK), 1 h, 3 h, 6 h, 12 h, and 24 h as compared to the transcript level at 0 h after salt treatment. (**c**–**g**) Analysis of KEGG enrichment pathways for DEGs under salt treatment for 1 h (**c**), 3 h (**d**), 6 h (**e**), 12 h (**f**), and 24 h (**g**). Rich factor is represented by the horizontal axis, and pathway name is indicated by the vertical axis. The size of the dots represents the number of genes in the pathway, and the color indicates *q*-value. Wolfberry seedings were exposed to the salt stress of 300 mM of NaCl for 0 h (CK), 1 h, 3 h, 12 h, and 24 h.

**Figure 9 genes-14-01943-f009:**
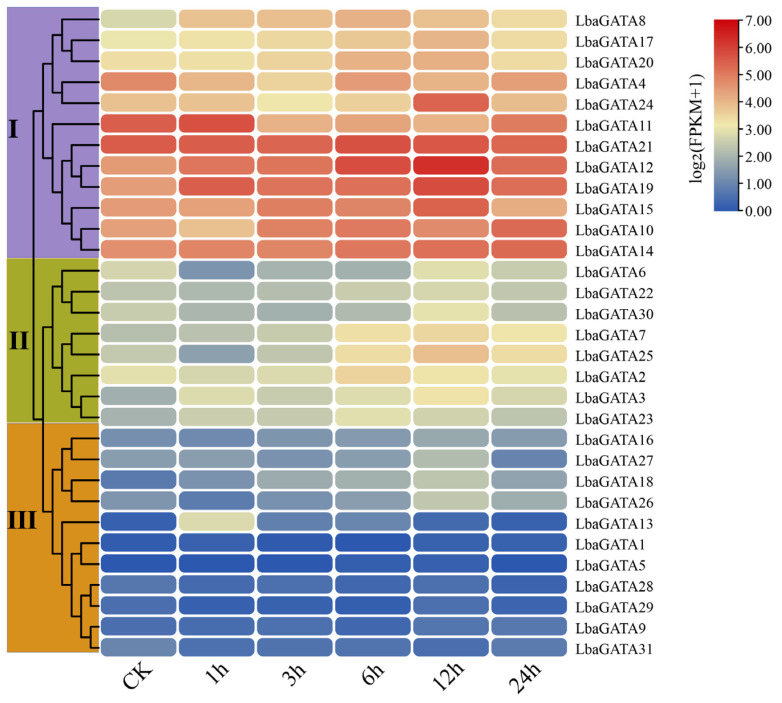
The heatmap of the 31 *LbaGATA* genes under salt stress using the TBtools v1.12. Wolfberry seedlings were exposed to the salt stress of 300 mM of NaCl for 0 (control, CK), 1, 3, 6, 12, and 24 h. Heatmaps were generated using the FPKM values of the 31 salt-stress-responsive *LbaGATAs* using the TBtools v1.12. The color scale beside the heatmap indicates gene expression levels, blue color indicates low transcript abundance, and red color indicates high transcript abundance.

**Figure 10 genes-14-01943-f010:**
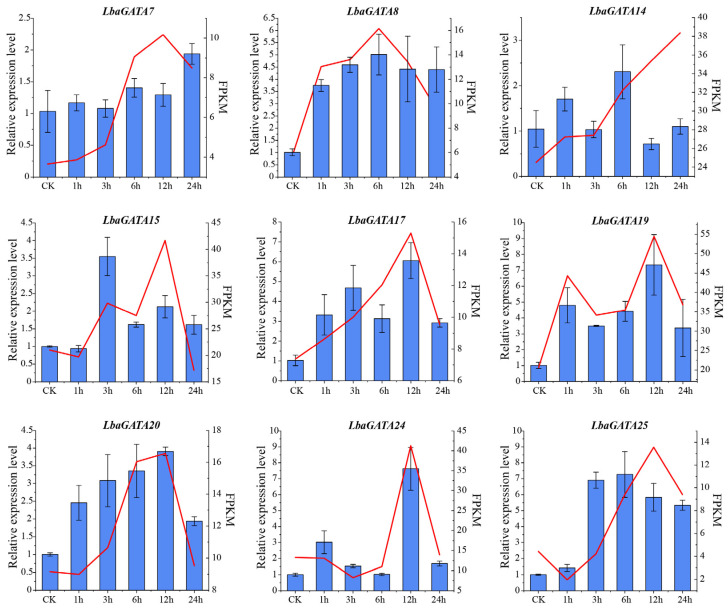
Gene expression patterns of nine *GATA* genes verified using qRT-PCR analysis under salt stress treatment. The left-hand y-axis represents the relative expression levels obtained using qRT-PCR analysis, and the right-hand y-axis represents the FPKM values ascertained using RNA-seq analysis. Wolfberry seedlings were exposed to the salt stress of 300 mM of NaCl for 0 h (CK), 1 h, 3 h, 6 h, 12 h, and 24 h. The qRT-PCR data were normalized against *LbaActin*, and the expression level at the first timepoint (0 h, CK) was set as 1. Three biological replicates are depicted with error bars.

**Table 1 genes-14-01943-t001:** Information on the genomes of five Solanaceae species and GATA gene numbers identified.

Common Name	Scientific Name	Chromosome Number (2n)	Genome Size	Genome Gene Number	*GATA* Genes	Gene Name Prefix
Wolfberry	*L. barbarum*	24	1.67 Gb	3,3581	31	Lba
Tomato	*S. lycopersicum*	24	785 Mb	3,4075	32	Sl
Pepper	*C. annuum*	24	3.3 Gb	3,5336	28	Ca
Potato	*S. tuberosum*	24	741.6 Mb	4,4851	36	St
Eggplant	*S. melongena*	24	1.07 Gb	3,6568	29	Sm

## Data Availability

The wolfberry (*L. barbarum* L.) genome sequence was downloaded from the NCBI database (https://www.ncbi.nlm.nih.gov/genome/?term=PRJNA640228, accessed on 15 September 2022). The *Solanum lycopersicum*, *Capsicum annuum*, *Solanum tuberosum*, and *Solanum melongena* genome sequences were downloaded from the Sol Genomics Network (https://solgenomics.net/, accessed on 10 October 2022) and Spud DB (http://spuddb.uga.edu/index.shtml), while the Arabidopsis GATA protein sequences were obtained from the TAIR website (https://www.arabidopsis.org/, accessed on 10 October 2022). All transcriptome data sets in this study have been uploaded to the NCBI (SRA) under the accession number PRJNA936477.

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
