# Peer review of "Comparative Analysis of the GATA Transcription Factors in Five Solanaceae Species and Their Responses to Salt Stress in Wolfberry (Lycium barbarum L.)"

_genes, 2023, doi:10.3390/genes14101943_

Round 1

Reviewer 1 Report

The main objective of this study is to elucidate the potential regulatory functions of LbaGATA genes in response to salt stress conditions. To comprehensively address this goal, a genome-wide investigation and comparative assessment of the GATA gene family, specifically its members, was conducted across a range of five different botanical species.  In addition, the research efforts included multifaceted analyses that included phylogenetic assessments, exon-intron structural evaluations, motif composition delineations, gene duplication investigations, and comprehensive assessments of cis-acting elements. In particular, an amalgamation of transcriptome analysis and quantitative real-time polymerase chain reaction (qRT-PCR) methods was employed to discern the specific responses of GATA family members to salt-induced stress within the wolfberry species.  The results of this study hold great promise for enhancing the collective understanding of the evolutionary dynamics underlying the GATA gene repertoire within the Solanaceae family. In addition, they provide valuable insights for the future pursuit of functional characterization of GATA genes not only in wolfberry, but also in other members of the Solanaceae family.

The introduction is presented correctly, in accordance with the subject. Numerous scientific articles, in concordance to the topic of the study, were consulted. The discussions are appropriate, in the context of the results, and was conducted compared to other studies in the field.

The following aspects are brought to the attention to the authors:

Methodology of the study was clearly presented, and appropriate to the proposed objectives with one exception The authors should also clearly specify whether replicates were included in the analyzed data or not? When starting analyzes to determine expression, a minimum of 3 replicates should be used in RNA-seq.

Author Response

Response to Reviewer 1 Comments

Point 1: Methodology of the study was clearly presented, and appropriate to the proposed objectives with one exception The authors should also clearly specify whether replicates were included in the analyzed data or not? When starting analyzes to determine expression, a minimum of 3 replicates should be used in RNA-seq.

Response 1:Thank you for your valuable advice. In this study, all data analyses included at least three biological replicates. We modified the method description in Line 166.

Reviewer 2 Report

In the manuscript “Comparative Analysis of the GATA Transcription Factors in Five Solanaceae Species and Their Responses to Salt Stress in Wolfberry (Lycium barbarum L.)”, the authors identified and characterized members of the GATA TF family in wolfberry, compare them to GATA TF previously identified in other four Solanaceae species and analyzed the expression of these genes under salt stress in wolfberry.

Data analyses were done well, but some details need to be clarified in the materials and methods section. Figures need improvement for the reader to be able to read genes and categories details. Overall, the manuscript needs minor improvement before publication in terms of grammar.

The minor issues found with the manuscript are described below:

·         Introduction (page 2, line 77): Authors indicate “In recently,” but this expression looks like a typo.

·         Materials and Methods (page 3, line 139): Authors indicate “intros” but this seems to be a typo and the correct word should be “introns”.

·         Materials and Methods (page 4, line 155): Authors indicate “ka”, but the correct expression would be “Ka”.

·         Materials and Methods (page 4, line 166 and 167): Authors indicate “The total RNA from the leaves collected at 0, 1, 3, 6, 12, and 24 h under salt stress was extracted using transcriptome data.” But this sentence does not make sense. Please indicate how was the RNA extracted (commercial kit, etc). Also, were the control samples (0 mM NaCl) analyzed? If a control was not analyzed, how do the authors identify DEGs in response to salt without a control? The authors indicate in a sentence that “the differentially expressed genes (DEGs) between the mapped sequences and the reference genome were identified using the DESeq2 v1.18 [49] with |log2(fold change)| ≥1 and false discovery rate <0.05 parameters.” But that does not make sense for identification of DEGs. DEGs are usually the ones found differentially expressed between a sample that was treated (salt treatment in this case) and a sample that was not (control conditions). Therefore, it is not clear how the authors identified the DEG under salt stress. Did the authors use 0 hours of salt stress as the control instead? If that was the case, that needs to be clarified in the methods.

·         Materials and Methods (page 4, lines 179 to 180): Authors indicate “RNA was extracted from leaves using Trizol reagent (TIANGEN, Beijing, China) following the manufacturer’s instructions.” Please clarify if samples used were the same as the ones used for the transcriptome analysis (0, 1, 3, 6, 12, and 24 h under salt stress). Was the control also analyzed?

·         Materials and Methods (page 4, line 185): Authors indicate “7.5 μL Mix” but never mentioned which was the name of the commercial master mix used for the qPCR. Also, was the Tm used for all primers the same? What was the reaction efficiency for each primer pair?

·         Results (page 5, line 206): Authors are missing a space between these words “SolanaceaeFamily”

·         Figure 1: Figure font needs to be bigger in size to be able to read the individual names of each of the genes in the tree. Or the figure needs to be bigger in size.

·         Figure 2a: Make the scale a Kb scale instead of a bp scale, so the numbers that are long are clearly separated than the rest of numbers to the left or the right on the scale.

·         Results (page 7, lines 258 and 259): Authors indicate “We used DupGen_finder software [45] to detect duplicated GATA family gene pairs in five Solanaceae genomes The WGD, TD, TRD, and”, but there seems to be missing a period “.” between sentences.

·         Figure 4. Figure font needs to be bigger in size to be able to read the individual names of each of the genes in the trees. Or the figure needs to be bigger in size.

·         Figure 7a. What does the colors (red to white) in each cell on the heatmap represent? Is that a color scale based on the number of cis elements found for each category in each gene promoter?

·         Figure 8. Figure font needs to be bigger in size to be able to read the individual names of each of the categories on the KEGG enrichment scatterplots. Or the scatterplots need to be bigger in size.

·         Figure 10. It would be good if all the graphs in composite figure 10 have the same y-axis scale (like 0 to 10) to be also able to compare the level of expression among all genes analyzed and realize easily which one has the highest expression in the group.

·         Discussion (page 14, line 411): Authors indicate “potatoes” instead of “potato”.

·         Discussion (page 15, line 439): Authors indicate “potatoes” instead of “potato”.

Few grammar mistakes and typos found.

Author Response

Response to Reviewer 2 Comments

Point1: Introduction (page 2, line 77): Authors indicate “In recently,” but this expression looks like a typo.

Response 1: We used “Recently” instead of “In recently”.

Point2: Materials and Methods (page 3, line 139): Authors indicate “intros” but this seems to be a typo and the correct word should be “introns”.

Response 2: We modified “intros” to “introns”.

Point3: Materials and Methods (page 4, line 155): Authors indicate “ka”, but the correct expression would be “Ka”.

Response 3: We modified this word.

Point4: Materials and Methods (page 4, line 166 and 167): Authors indicate “The total RNA from the leaves collected at 0, 1, 3, 6, 12, and 24 h under salt stress was extracted using transcriptome data.” But this sentence does not make sense. Please indicate how was the RNA extracted (commercial kit, etc). Also, were the control samples (0 mM NaCl) analyzed? If a control was not analyzed, how do the authors identify DEGs in response to salt without a control? The authors indicate in a sentence that “the differentially expressed genes (DEGs) between the mapped sequences and the reference genome were identified using the DESeq2 v1.18 [49] with |log2(fold change)| ≥1 and false discovery rate <0.05 parameters.” But that does not make sense for identification of DEGs. DEGs are usually the ones found differentially expressed between a sample that was treated (salt treatment in this case) and a sample that was not (control conditions). Therefore, it is not clear how the authors identified the DEG under salt stress. Did the authors use 0 hours of salt stress as the control instead? If that was the case, that needs to be clarified in the methods.

Response 4: In this study, 0 hour of salt stress as the control (CK). We modify this part of the method as following:

Materials and Methods: Lines 97 to 105 (Wolfberry (L. barbarum cv Ningqi 1) seeds were sown on moist filter papers on Petri plates for germination. Next, the seedlings were transplanted into a plastic box containing 1/2 Hoagland solution and maintained for four weeks later. Subsequently, the seedlings were subjected to salt treatments. Briefly, the seedlings were transferred into a hydroponic culture consisting of Hoagland solution under 300 mM NaCl treatment and incubated at 22/18 °C (day/night), 65% relative humidity, and 16 h photoperiod. Subsequently, the leaves tissues were collected at 0 h (samples without treatment, CK), 1 h, 3 h, 6 h, 12 h, and 24 h after salt application, grounded, frozen in liquid nitrogen, and stored at -80 °C awaiting RNA-Seq and qRT-PCR analyses. Each treatment had at least six seedlings and three biological replicates.

Materials and Methods: (Lines 165 to 168): The ground-above tissues at 0, 1, 3, 6, 12, and 24 after salt stresses treatment were collected and applied to RNA-Seq analysis. Total RNA was extracted from treatment samples using Trizol method as described previously [12] and mRNA libraries were sequenced on the Illumina HiSeq4000 platform.

Point5: Materials and Methods (page 4, lines 179 to 180): Authors indicate “RNA was extracted from leaves using Trizol reagent (TIANGEN, Beijing, China) following the manufacturer’s instructions.” Please clarify if samples used were the same as the ones used for the transcriptome analysis (0, 1, 3, 6, 12, and 24 h under salt stress). Was the control also analyzed?

Response 5: Yes, the samples used for qRT-PCR and transcriptome analysis are the same samples. The control was also analyzed. We modified as following: The expression patterns of the LbaGATA genes were analyzed by qRT-PCR. Total RNA was extracted from leaves (the sample same as transcriptome analysis) using Trizol reagent (TIANGEN, Beijing, China) following the manufacturer’s instructions.

Point6: Materials and Methods (page 4, line 185): Authors indicate “7.5 μL Mix” but never mentioned which was the name of the commercial master mix used for the qPCR. Also, was the Tm used for all primers the same? What was the reaction efficiency for each primer pair?

Response 6: We added the name of the commercial master mix, ..7.5 μL Mix (TransGen, Beijing, China)..

Point7: Results (page 5, line 206): Authors are missing a space between these words “SolanaceaeFamily”

Response 7: We added a space between the words “SolanaceaeFamily”.

Point8: Figure1 Figure font needs to be bigger in size to be able to read the individual names of each of the genes in the tree. Or the figure needs to be bigger in size.

Response 8: Thank you for your valuable advice. We modified the font size in Figure 1.

Point 9: Figure 2a: Make the scale a Kb scale instead of a bp scale, so the numbers that are long are clearly separated than the rest of numbers to the left or the right on the scale.

Response 9: We made the scale a Kb scale instead of a bp scale in Figure 2b.

Point 10: Results (page 7, lines 258 and 259): Authors indicate “We used DupGen_finder software [45] to detect duplicated GATA family gene pairs in five Solanaceae genomes The WGD, TD, TRD, and”, but there seems to be missing a period “.” between sentences.

Response 10: To investigate the origin of GATA family genes, we used DupGen_finder software to detect duplicated GATA family gene pairs in five Solanaceae genomes. The WGD, TD, TRD, and DSD duplication events were present in all the five species from the Solanaceae family, while PD duplication event was traced only in tomato, pepper, and potato.

Point 11: Figure 4. Figure font needs to be bigger in size to be able to read the individual names of each of the genes in the trees. Or the figure needs to be bigger in size.

Response 11: We modified the font size in Figure 4.

Point 12: Figure 7a. What does the colors (red to white) in each cell on the heatmap represent? Is that a color scale based on the number of cis elements found for each category in each gene promoter?

Response 12: We added color scar on right Figure 7a. The scale represents number of cis-elements.

Point 13: Figure 8. Figure font needs to be bigger in size to be able to read the individual names of each of the categories on the KEGG enrichment scatterplots. Or the scatterplots need to be bigger in size.

Response 13: We modified the font size in Figure 8.

Point 14: Figure 10. It would be good if all the graphs in composite figure 10 have the same y-axis scale (like 0 to 10) to be also able to compare the level of expression among all genes analyzed and realize easily which one has the highest expression in the group.

Response 14: We adjusted the y-axis scale to 0 to 10, and the picture looks out of place.

Point 15: Discussion (page 14, line 411): Authors indicate “potatoes” instead of “potato”.

Response 15: We modified “potatoes” to “potato”.

Point 16: Discussion (page 15, line 439): Authors indicate “potatoes” instead of “potato”.

Response 16: We modified “potatoes” to “potato”.

Reviewer 3 Report

The current research conducted by Zhang et al. is quite impressive and also quite extensive. In my opinion, the manuscript is well-written too. However, I have some suggestions, that the authors might consider prior to the final acceptance.

1. The authors chose 5 Solanaceae species, among them, 3 are from the genus “Solanum”. This might produce biased results.

2. Please include the bootstrap values in the phylogenetic tree.

3. The authors should consider including CpG analysis of the promoter sequences.

4. In the Figure 8 caption, 0 h is written, but the results for the 0 h is not shown in the figure.

5. Authors should consider doing co-evolution analysis (bioinformatic analysis) and 3D-structure modeling to get more comprehensive insights into the GATA Transcription Factor families.

Author Response

Point 1: The authors chose 5 Solanaceae species, among them, 3 are from the genus “Solanum”. This might produce biased results.

Response 1:Thank you for your good idea. Wolfberry belong to Solanaceae family along with tomato, eggplant and potato. Wolfberry (Lycium barbarum L.) is an important Chinese traditional herbal medicine, which is mainly grown in northwest China, where there is serious soil salinity. The GATA proteins are a class of zinc-finger DNA-binding proteins that participates in diverse regulatory processes in plants, especially in response to abiotic stress. However, the GATA genes related to salt stress in L. barbarum have not been identified. This study aimed to explore the potential regulatory roles of LbaGATA genes in response salt stress.

Point 2: Please include the bootstrap values in the phylogenetic tree.

Response 2:In Figure 1and 2, purple dot represents bootstrap values. The size of the point represents the value big or small.

Point 3: The authors should consider including CpG analysis of the promoter sequences.

Response 3:Thank you for your valuable advice. In the further study, we should consider including CpG analysis of the promoter sequences.

Point 4: In the Figure 8 caption, 0 h is written, but the results for the 0 h is not shown in the figure.

Response 4:In this study, 0 h (sample without treatment, CK), hence, 0 h is not shown in the figure.

Point 5: Authors should consider doing co-evolution analysis (bioinformatic analysis) and 3D-structure modeling to get more comprehensive insights into the GATA Transcription Factor families.

Response 5:Thank you for your valuable advice. In this study, we found that four candidate LbaGATA genes (LbaGATA8, LbaGATA19, LbaGATA20, and LbaGATA24) are potentially involved in salt stress response. In the further study, we should consider 3D-structure modeling to know these genes’ function.

Round 2

Reviewer 1 Report

The article is ready for publication.